

# Atmospheric turbulence affects wind turbine nacelle transfer functions

Clara M. St. Martin,[1] Julie K. Lundquist,[1,2] Andrew Clifton,[2] Gregory S. Poulos,[3] and Scott J.

Schreck[2]

[1] Department of Atmospheric and Oceanic Sciences (ATOC), University of Colorado at Boulder, 311 UCB, Boulder, CO, 80309

[2] National Renewable Energy Laboratory, 15013 Denver West Parkway, Golden, CO, 80401

[3] V-Bar, LLC, 1301 Arapahoe Street, Suite 105, Golden, CO, 80401

Correspondence to: Clara M. St. Martin (clara.st.martin@colorado.edu)





**Abstract.** Despite their potential as a valuable source of individual turbine power performance and turbine array energy production optimization information, nacelle-mounted anemometers have often been neglected because complex flows around the blades and nacelle interfere with their measurements. This work quantitatively explores the accuracy of and potential corrections to nacelle anemometer measurements to determine the degree to which

they may be useful when corrected for these complex flows, particularly for calculating annual energy production (AEP) in the absence of other meteorological data. Using upwind meteorological tower measurements along with nacelle-based measurements from a General Electric (GE) 1.5sle model, we calculate empirical nacelle transfer functions (NTFs) and explore how they are impacted by different atmospheric and turbulence parameters. This work provides guidelines for the use of NTFs for deriving useful wind measurements from nacelle-mounted anemometers.

Corrections to the nacelle anemometer wind speed measurements can be made with NTFs and used to calculate an AEP that comes within 1 % of an AEP calculated with upwind measurements.  We also calculate unique NTFs for different atmospheric conditions defined by temperature stratification as well as turbulence intensity, turbulence kinetic energy, and wind shear. During periods of low stability as defined by the Bulk Richardson number ($R_B$), the nacelle-mounted anemometer underestimates the upwind wind speed more than during periods of high stability at

some wind speed bins below rated speed, leading to a more steep NTF during periods of low stability. Similarly, during periods of high turbulence, the nacelle-mounted anemometer underestimates the upwind wind speed more than during periods of low turbulence at most wind bins between cut-in and rated wind speed. Based on these results, we suggest different NTFs be calculated for different regimes of atmospheric stability and turbulence for power performance validation purposes.

**Keywords**

nacelle anemometry, nacelle transfer function, atmospheric stability, turbulence





## 1 Introduction

Traditionally, each wind turbine has an anemometer and wind vane mounted on its nacelle, behind the hub (Fig.1).

Measurements collected from these instruments are used for yaw control and turbine cut-in/cut-out procedures.

Nacelle measurements could also be used to help improve turbine or park efficiency. For example, power

performance verifications for individual turbines can now be based on the nacelle anemometer with suitable nacelle

transfer functions (NTFs) (International Electrotechnical Commission [IEC] 61400-12-2 2013). Nacelle

measurements can also provide critical input for wind farm production optimization (Fleming et al., 2016). With

sufficiently accurate NTFs, these data can provide a valuable, extensive, and continuous source of turbine-specific

performance information.

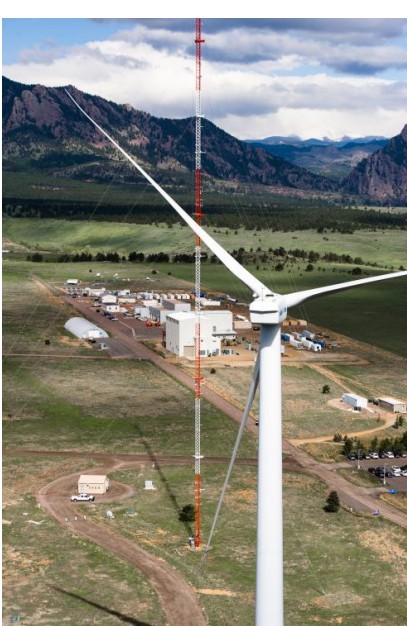

**Figure 1**. GE-1.5/77 sle turbine at the National Wind Technology Center. Photo credit: Dennis Schroeder/NREL (image gallery number 25872).

Power performance validation has traditionally relied on hub-height wind speed observations from a

meteorological (met) tower upwind of a turbine (Link and Santos, 2004; IEC 61400-12-1, 2015). The IEC 61400-

12-1 (2015) standards require a met tower to be placed at the turbine location prior to turbine erection (the so-called





"site calibration" procedure) for a power performance test to be considered valid (of sufficiently low total

uncertainty) in complex terrain. However, it is not feasible to erect "site calibration" met towers after the turbine has

been erected. And, even if "site calibration" is not required because a site is in simple terrain, tower erection is time-

consuming and unrealistic to complete for every turbine at a given park. These factors motivate exploration of the

use of nacelle-mounted anemometers to provide wind speed data for power performance validation.

Several studies have found that nacelle anemometer measurements can be adjusted by the use of transfer

functions between some upwind hub-height measurement and the nacelle-mounted anemometer measurement

(Antoniou and Pedersen, 1997; Hunter et al., 2001, Smith et al., 2002; Smaïli and Masson, 2004). The IEC 61400-

12-2 (2013) standard now allows the use of nacelle-mounted anemometers to verify power curves based on these

transfer functions, or fitted functions of correction factors between upwind hub-height wind speed (UHWS)

measurements and nacelle-mounted anemometer wind speed (NAWS) measurements.

An empirical NTF may not result in a linear relationship between the UHWS and NAWS. In fact, Antoniou

and Pedersen (1997) found that the transfer functions fit well with a fifth-order polynomial curve. Hunter et al.

(2001) similarly found a non-linear relationship and that a linear regression would overestimate the wind speed

between 6 and 11 m s$^{-1}$ and underestimate the wind speed at wind speeds less than 4 m s$^{-1}$ and greater than 16 m s$^{-1}$.

Smith et al. (2002) found a linear relationship with the exception of wind speeds below cut-in and wind speeds about

15 m s$^{-1}$.

In previous work, the relationship between UHWS measurements and NAWS measurements has been

found to depend on multiple factors. Antoniou and Pedersen (1997) found that relations between the UHWS and the

NAWS were dependent on rotor settings such as blade pitch angle and the use of vortex generators, yaw error,

anemometer position, and terrain. They concluded that if these factors were kept constant, the relation could be used

for all wind turbines of the same make and type. Frandsen et al. (2009) found a dependence on flow induction

caused by the rotor. Dahlberg et al. (1999) discovered that pitch angle affects the relation. Dahlberg et al. (1999),

Smith et al. (2002), and Frandsen et al. (2009) also stressed the importance of the correct calibration of the nacelle

anemometers and that this calibration has an effect on the error measured in the relation. Zahle and Sørensen (2011)

found that the inflow angle to the rotor and yaw misalignment influences the relationship. Smith et al. (2002)

concluded the relation may depend on turbine controls, topography, and nacelle height and position. Smaïli and




Masson (2004) implemented a numerical model and concluded that a relation should account for rotor-nacelle interactions and hypothesized that wakes, topography, and nacelle misalignment would all have some effect on the relation. To summarize, the factors found to be relevant in NTFs are: rotor settings, yaw error, anemometer position,

terrain, flow induction (decrease in wind speed just in front of or just behind the rotor), nacelle anemometer calibration, and inflow angle.

The roles of inflow turbulence and atmospheric stability on NTFs have not yet been explored. However, previous work on power performance and annual energy production (AEP) does acknowledge the role of atmospheric stability, wind shear, and turbulence intensity (TI) in inducing deviations of power from the

manufacturer power curve (MPC) (e.g., Sumner and Masson, 2006; Antoniou et al., 2009; Rareshide at el., 2009; Wagenaar and Eecen, 2011; Wharton and Lundquist, 2012; Vanderwende and Lundquist, 2012; St. Martin et al., 2016).

In this study, we quantify the effect of NTF-corrected nacelle anemometer measurements on the AEP and investigate the influence of different atmospheric stability and turbulence regimes on these NTFs. In Sect. 2, we

briefly summarize our data set, which includes upwind as well as nacelle-based measurements, as well as our data analysis methods which include filtering based on turbine operation, and definitions of the stability and turbulence regimes. We present results of AEP calculations as well as separate NTFs for different stability and turbulence regimes in Sect. 3, and in Sect. 4 we summarize conclusions about the effect of the NTF on the AEP as well as the effects of atmospheric stability and inflow turbulence on the NTFs.

**2 Data and methods**

**2.1 Meteorological and turbine data**

For this analysis, we use 2.5 months of data collected at the U.S. Department of Energy (DOE) National Wind Technology Center (NWTC) at the National Renewable Energy Laboratory (NREL) during the wintertime (29 November 2012–14 February 2013). Ten-minute-averaged turbine supervisory control and data acquisition

(SCADA) data used in this study are from a General Electric (GE)-1.5-MW turbine (GE-1.5/77 sle, Fig.1), with a cut-in wind speed of 3.5 m s$^{-1}$, rated wind speed of 14 m s$^{-1}$, and cut-out wind speed of 25 m s$^{-1}$. A map of the site



can be found in St. Martin et al. (2016) (Fig. 1). See Mendoza et al. (2015) for power performance test results from the DOE GE-1.5 along with instrument and site calibration information.

Upwind data include 1-Hz measurements of wind speed and direction averaged to 10 min from a
Renewable NRG Systems (NRG)/LEOSPHERE WINDCUBE v1 vertically profiling Doppler lidar (2.7 D upwind) and 10- and 30-min averages from a 135-m met tower (2.0 D upwind). Volumetric-averaged wind speeds and directions are measured by the lidar every 20 m, from 40 m to 220 m. Comparison of the lidar wind profiles to those from the met tower suggest that the lidar data at this site suffered from inhomogeneities as a result of complex flows (Bingöl et al., 2009; Rhodes and Lundquist, 2013; Lundquist et al., 2015), and so the majority of this paper will
focus on the results of the analysis using the tower data. On the met tower, cup anemometers placed at 3, 10, 30, 38, 55, 80, 87, 105, 122, and 130 m measure wind speed, vanes placed at 3, 10, 38, 87, and 122 m measure wind direction, and three-dimensional (3-D) sonic anemometers placed at 15, 41, 61, 74, 100, and 119 m measure the components of the wind. Barometric pressure and precipitation amounts are measured at 3 m and temperature is measured at 3, 38, and 87 m. See Fig. 2 in St. Martin et al. (2016) for a schematic of the tower.

As discussed in St. Martin et al. (2016), meteorological and turbine data are filtered for quality assurance. Data are only considered during time periods when the turbine is operating and wind direction indicates the turbine is located downwind of the lidar and met tower (235°–315°). As the turbine data from the SCADA system is available in 10-min increments, variability of the turbine parameters on a shorter timescale cannot be discerned. However, we filter for "normal turbine operation" based on curtailment using generator speed set point for wind
speeds greater than 5.5 m s$^{-1}$, whereas for wind speeds less than 5.5 m s$^{-1}$, we discard data when the turbine is not grid-connected and is faulted (Fig. 6 in St. Martin et al., 2016).

Further, it is possible that the nacelle-reported wind speeds used in this analysis have been subjected to a simple, built-in transfer function before the retrieval from the SCADA system of the DOE GE 1.5sle turbine. We see this uncertainty as an advantage to our analysis as a typical wind plant operator would only have access to similar
data.

**2.2 AEP calculations**



To simulate a scenario in which a wind plant operator only has nacelle-based measurements and no upwind tower or remote-sensing measurements, we calculate an AEP (as described in Sect. 9.3 of IEC 61400-12-2, 2013) using only nacelle winds to compare to an AEP calculated with upwind met tower 80-m winds. We then correct the nacelle-

based measurements with NTFs and calculate AEP based on these results for comparison as well. Although data for this analysis only spans 2.5 months in the wintertime at the NWTC during the 2012–2013 season, we calculate AEPs using the total amount of hours in an entire year to show values close to a representative AEP value. A sample wind distribution using Weibull distribution parameters representative of the data set (scale parameter: $\lambda = 10.04$ m s$^{-1}$, shape parameter: $k = 2.63$, figure not shown) is used in these calculations as suggested by IEC 61400 12-1

(2015) for a site-specific AEP.

### 2.3 Stability metrics

We calculate Bulk Richardson number ($R_B$), Obukhov length ($L$), and the power law exponent ($\alpha$) and use these as stability metrics for these data. Using wind speed and temperature differences between surface and upper tip (3 m and 122 m, respectively) tower measurements, we calculate 10-min values of $R_B$ to compare the buoyant production

of turbulence to the mechanical production of turbulence. Using near-surface flux measurements at 15 m (within the surface layer) as well as surface temperature and humidity measurements interpolated to 15 m, we calculate 30-min values of $L$ to estimate the height at which the buoyant production of turbulence dominates the mechanical production of turbulence. Using horizontal wind speeds as measured by cup anemometers at 38 m and 122 m (lower tip and upper tip of the rotor disk), we calculate 10-min values of $\alpha$ to quantify the wind speed vertical profile across

the rotor disk. Though some previous studies combine metrics to define stability (Vanderwende and Lundquist, 2012), the three atmospheric stability metrics discussed here are treated separately with regard to the NTFs because of slight differences between their definitions of unstable and stable conditions (see Fig. 11 in St. Martin et al., 2016). These differences may be attributed to distinctions between each approach in defining atmospheric stability, a difference in averaging period, heights of the measurements used in the calculations, or how $R_B$ and $L$ use wind

speed and temperature measurements to define stability, whereas $\alpha$ uses only wind speed measurements.

Further, we calculate TI and turbulence kinetic energy (TKE) to provide turbulence metrics and estimate the effect of hub-height inflow turbulence on the NTFs. Using 80-m wind speed measurements from the met tower, we calculate 10-min values of TI, or the standard deviation of the horizontal wind speed normalized by the average





horizontal wind speed at hub height. Using 74-m wind measurements from a 3-D sonic anemometer on the tower,

we calculate 30-min values of TKE per unit mass, or the sum of the variances of the components of the wind divided

by two. Note that after filtering out spikes in the raw 74-m sonic anemometer data, only about 367 thirty-min TKE

values remain (183.5 h) and the fewer number of data points likely affects the statistical significance of the NTFs for

different TKE regimes.

Regimes or classifications for these stability and turbulence parameters are defined in Table 1 and

described in detail in St. Martin et al. (2016), along with more detailed descriptions of the data from the lidar, tower

and turbine, as well as filtering methods.

**Table 1.** Defined stability and turbulence regimes.

| Regime | $R_B$ | $L$ (m) | $\alpha$ | TI (%) | TKE ($m^2\,s^{-2}$) |
|--------|-------|---------|----------|--------|---------------------|
| **Low** | $R_B < -0.03$ | $-1{,}000 < L \leq 0$ | $\alpha < 0.11$ | TI < 15 | TKE < 3.0 |
| **Medium** | $-0.03 < R_B < 0.03$ | $0 \leq L < 1{,}000$ | $0.11 < \alpha < 0.17$ | 15 < TI < 20 | 3.0 < TKE < 6.5 |
| **High** | $R_B > 0.03$ | $|L| \geq 1{,}000$ | $\alpha > 0.17$ | TI > 20 | TKE > 6.5 |

## 3 Results

To explore the variability of the NTF, we calculate specific NTFs filtered by atmospheric stability metrics, TI, and

TKE. We investigate filters that have either previously been found to affect the transfer function or are suspected to

have an effect on the transfer functions based on power curve studies (e.g., St. Martin et al., 2016). Additionally, we

explore the effects of yaw error and wind veer and distributions of these variables, but, as in St. Martin et al. (2016),

yaw error and wind veer do not seem to impact either the power curves or the NTFs at this site and therefore are not

shown.

### 3.1 Preliminary NTFs

A general NTF (Fig. 2a) compares the tower 80-m wind speed to the nacelle-reported wind speed using all data that

pass the wind speed, wind direction, and normal operation criteria defined in Sect. 2.1 and in more detail in Sect. 3.2

and 3.3 in St. Martin et al. (2016). As a fifth-order polynomial fit was found to be suitable for power curve

assessment in previous work by Antoniou and Pedersen (1997) and Hunter et al. (2001), we also apply this type of




fit to the wind speeds in this work to estimate an empirical transfer function between 80-m tower wind speed

measurements and nacelle-mounted anemometer wind speed measurements (Fig. 2a). The r-squared value of the

fifth-order polynomial fit to the data is 0.9912, which means the fit line predicts 99.12% of the variance in the tower

data. The root-mean-square error (RMSE) of the fifth-order polynomial fit is 0.3615 m s$^{-1}$. After correcting the

nacelle-measured wind speeds using this NTF, deviations between the corrected nacelle wind speed and the tower

80-m wind speeds (Fig. 2b) vary between -0.2 and 0.2 m s$^{-1}$ throughout all wind speed bins between cut-in and cut-

out wind speed.

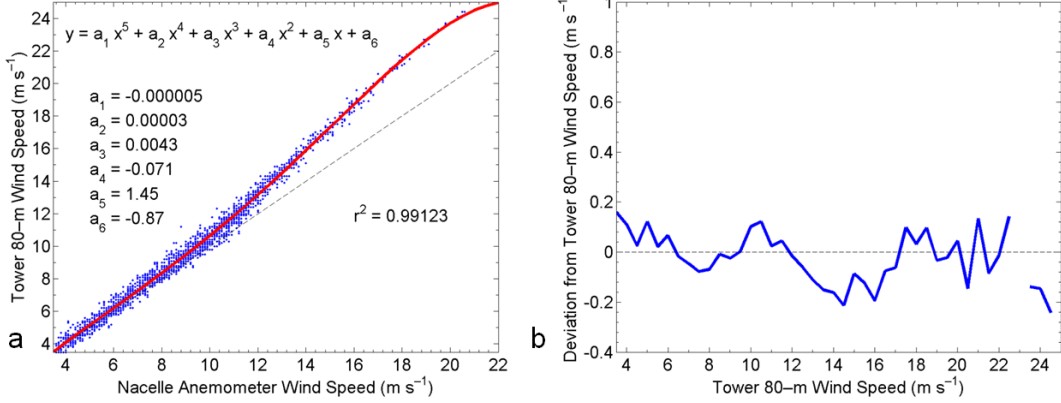

**Figure 2**. Comparison of upwind wind speeds with nacelle anemometer wind speeds: (a) Scatter is the upwind tower 80-meter

(m) wind speed versus nacelle wind speed. Red line is the fifth-order polynomial fit and empirical transfer function between the

tower 80-m observations and the nacelle-mounted anemometer observations. Dashed line is 1:1.  (b) Average deviation in fifth-

order polynomial nacelle transfer function (NTF)-corrected nacelle-mounted anemometer wind speed from tower 80-m wind

speed versus tower 80-m wind speed. The dashed line indicates a 0 m s$^{-1}$ change. The figure includes data filtered for the tower

80-m wind speeds between 3.5 and 25.0 m s$^{-1}$, 87-m wind directions between 235° and 315°, and for normal turbine operation.

Based on the small coefficients of the third-, fourth-, and fifth-order of the fit in Fig. 2a, a fifth-order

polynomial may be unnecessarily complex. Therefore, a second-order polynomial fit is also calculated to estimate an

empirical transfer function. The r-squared value of the second-order polynomial fit with the data is also very high,

0.9909 (Fig. 3a). The RMSE of the second-order polynomial fit is 0.3680 m s$^{-1}$. After correcting the nacelle-

measured wind speeds using this NTF, deviations between the corrected nacelle wind speed and the tower 80-m



wind speeds, shown in Fig. 3b, vary from about -0.3 to 0.2 m s$^{-1}$ at wind speed less than about 22 m s$^{-1}$ but grow to about 0.8 m s$^{-1}$ at higher wind speeds. Though there are fewer data points at these higher wind speed bins, this larger

deviation of the second-order NTF-corrected wind speeds from the upwind wind speeds at higher wind speeds suggests that a fifth-order polynomial NTF is unnecessary until high wind speeds are considered.

Both transfer functions for this dataset (Fig. 2a, Fig. 3a) are close to linear at low wind speeds but non-linear just before rated speed (14 m s$^{-1}$), hence the higher-order polynomial fits. This behavior suggests that at wind speeds below about 9 m s$^{-1}$, the nacelle anemometer measurement closely corresponds to the upwind wind speed.

Above this wind speed threshold, however, the nacelle anemometer underestimates the upwind wind speed by almost 2 m s$^{-1}$ around rated speed to about 4 m s$^{-1}$ at upwind wind speeds near 20 m s$^{-1}$;  higher ambient wind speeds are associated with more significant slow-downs around the nacelle.

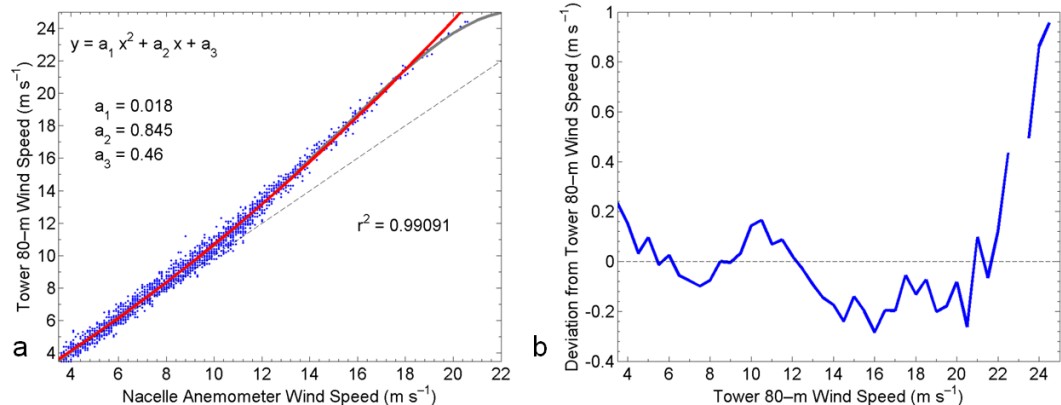

**Figure 3.** Comparison of upwind wind speeds with nacelle anemometer wind speeds. (a) Scatter is the upwind tower 80-m wind speed versus nacelle wind speed. Red line is the second-order polynomial fit and empirical transfer function between the tower 80-m observations and the nacelle-mounted anemometer observations; gray line is the fifth-order polynomial fit. Dashed line is 1:1. (b) Average deviation in the second-order polynomial NTF-corrected nacelle-mounted anemometer wind speed from tower 80-m wind speed versus tower 80-m wind speed is shown. Dashed line indicates a 0 m s$^{-1}$ change. Includes data filtered for tower 80-m wind speeds between 3.5 and 25.0 m s$^{-1}$, 87-m wind directions between 235° and 315°, and for normal turbine operation.



Comparison of the NTF developed from the upwind tower measurements and the NTF developed from the upwind lidar measurements (Fig. 4a) emphasizes that the lidar measurements exhibit greater variability ranging over all relevant wind speeds. The variability in the lidar measurements caused by the inhomogeneity of the flow suggests that the tower measurements are more reliable for calculating power curves and transfer functions at this particular site, which is known to experience complex and inhomogeneous flow (Aitken et al., 2014). Despite the larger variability in the lidar data set for both the transfer function (Fig. 4a) and deviations between the corrected

nacelle wind speed and the upwind wind speeds (Fig. 4b), both transfer functions in Fig. 4a show linearity at lower wind speeds and non-linearity at higher wind speeds.

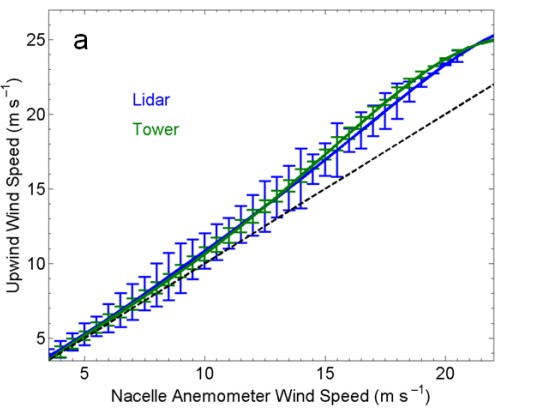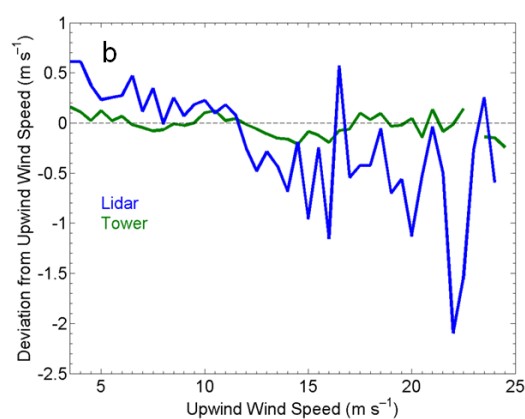

**Figure 4**. (a) NTFs calculated employing fifth-order polynomial fits using tower hub-height data (green) and lidar hub-height data (blue). Envelopes represent $\pm \sigma$ of the data within the same bins as the bins the NTFs are calculated with. Includes data filtered for tower 80-m wind speeds between 3.5 and 25.0 m s$^{-1}$, 87-m wind directions between 235° and 315°, and for normal turbine operation. Dashed line is 1:1; (b) shows the average deviation in NTF-corrected nacelle-mounted anemometer wind speed from tower 80-m wind speed (green) and lidar 80-m wind speed (blue) versus tower 80-m wind speed. Dashed line indicates a 0 m s$^{-1}$ change.

To try to quantitatively explain this change in the transfer function from linear to non-linear and to connect with possible flow blockage behind the rotor and along the nacelle, the non-dimensional Froude number (Stull,

1988) for flow around the nacelle is calculated. Froude numbers are investigated during stable conditions using measurements from the tower at the surface and around hub height and using a range of length scales from 1−10 m




to represent the length and width of the nacelle. However, distinctions between these two wind speed regions could not be seen in these calculations as Froude numbers were found to be positive and increase with increasing wind speed.

Additionally, because the transfer functions become non-linear between cut-in wind speed and rated speed, the transfer function may be impacted by turbine operations in that region of the power curve possibly because of root vortices (Whale et al., 2000). Just below rated speed, the blades begin to pitch forward to maintain rated generator speed, thus allowing power production to remain near rated power (Fig.5). This "feathering" of the blades changes the flow around the blades and therefore the wind that affects the nacelle-mounted anemometer

measurement. Though this hypothesis cannot be further investigated within this campaign as higher resolution data from the SCADA system are unavailable, this does make a compelling argument for installing 3-D sonic anemometers on nacelles so vertical velocity can be measured to further understand the 3-D wind structures behind the rotor and along the nacelle, and how these flow structures change as inflow wind speed increases.

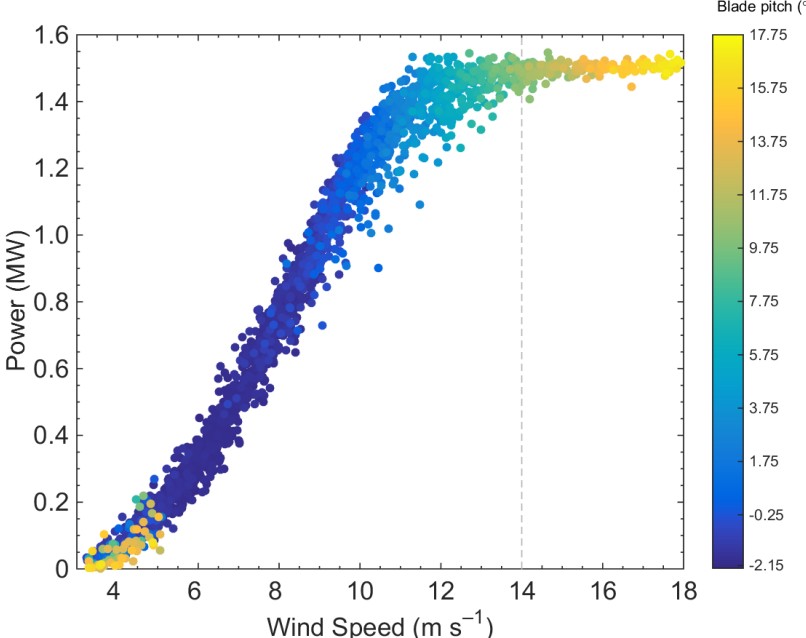

**Figure 5**. Scatter power curve using 80-m tower winds after filtering for wind speeds between 3.5 and 25 m s$^{-1}$, wind directions between 235°and 315°, and for normal turbine operation. Colors of the scatter points correspond to blade pitch angles.  The grey dashed line marks rated speed.



### 3.2 Annual energy production and NTFs

It is important to understand the characteristics of the NTF and how it changes with wind speed, as this under-

estimation of the ambient wind speed, especially at wind speeds in which the growth in power production with wind

speed is the most significant, could result in a significant overestimation of AEP in power performance verification.

With no NTF correction applied (aside from the transfer function that is built into the SCADA system by

the manufacturer), the AEP calculated with nacelle winds (AEP_nacelle) overestimates the AEP calculated with 80-

m tower winds (AEP_upwind) by 5.96 % (Table 2). This overestimation of AEP is expected as the nacelle

anemometer consistently underestimates the upwind wind speed, which leads to the misrepresentation of higher

power output at lower wind speeds and therefore a higher AEP.

**Table 2.** Top row: Annual energy production (AEP) in megawatt-hours/year calculated using upwind tower measurements

(AEP_upwind), nacelle winds (AEP_nacelle), corrected nacelle winds using the NTF calculated with a fifth-order polynomial

(AEP_NTF5$^{th}$), and corrected nacelle winds using the NTF calculated with a second-order polynomial (AEP_NTF2$^{nd}$). Bottom

row: AEP in percentage calculated as the difference from AEP_upwind.

|  | AEP_upwind | AEP_nacelle | AEP_NTF5$^{th}$ | AEP_NTF2$^{nd}$ |
|---|---|---|---|---|
| **AEP (MWh/y)** | 7,479.3 | 7,924.7 | 7,479.1 | 7,465.8 |
| **% difference from tower winds** | 100.00 | 105.96 | 100.00 | 99.82 |

The use of the general NTF to correct the nacelle anemometer measurements reduces the AEP error

significantly (Table 2). With the application of the fifth-order polynomial NTF (AEP_NTF5$^{th}$), AEP_NTF5$^{th}$

underestimates AEP_upwind by only 0.003 %, whereas with the application of the second-order polynomial NTF

(AEP_NTF2$^{nd}$), AEP_NTF2$^{nd}$ underestimates AEP_upwind by 0.18 %. Therefore, using either the fifth-order

polynomial or the second-order polynomial for the NTF results in an AEP similar to that of an AEP calculated with

upwind hub-height winds; though both lead to a slight underestimation.

### 3.3 Atmospheric stability effects of NTFs

The value of atmospheric-stability segregation for NTFs seems to depend on how stability is defined. Some

statistically significant distinctions in the NTFs for $R_B$-defined unstable and stable cases do emerge (Fig. 6a, Table




3), particularly for wind speeds between 7 and 11 m s$^{-1}$. Closed circles in Fig. 6a-c represent statistically distinct wind speed bins between the stability classes and are determined by the Wilcoxon rank sum test with a 1 %

significance level. Stable cases follow a linear relationship more closely for low and moderate wind speeds (less than 11 m s$^{-1}$), whereas unstable cases show more deviation from the 1:1 line at wind speeds greater than 8 m s$^{-1}$. Conversely, no statistically significant distinctions emerge in the NTFs for $L$-defined stability classes for this site using our dataset (Fig. 6b, Table 3). Distinctions in the NTFs for $\alpha$-defined cases (Fig. 6c, Table 4) emerge only around 13.5 m s$^{-1}$—much closer to rated speed—and stable cases underestimate the upwind wind speed more than

unstable cases.

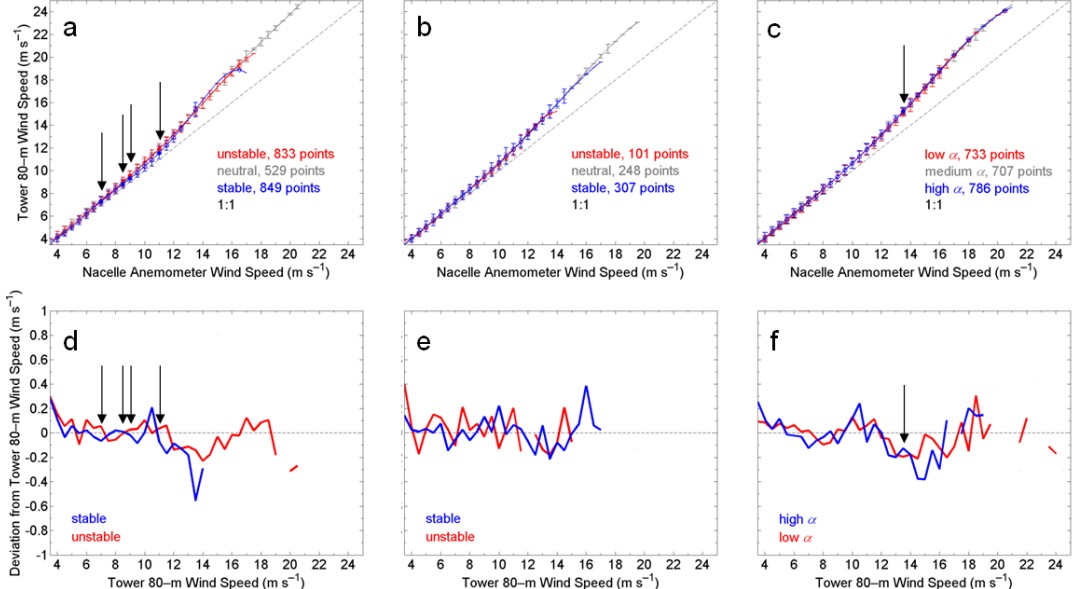

**Figure 6**. Tower 80-m NTFs calculated using fifth-order polynomial fits with stability regimes based on (a) $R_B$, (b) $L$, and (c) $\alpha$. Error bars represent ± σ of the data within the same bins as the bins with which the NTFs are calculated. Statistically distinct differences within each wind speed bin between the stability classes are determined by the Wilcoxon rank sum test with a 1 % significance level and denoted by closed circles. Black arrows point towards statistically distinct bins. The figures include data filtered for tower 80-m wind speeds between 3.5 and 25.0 m s$^{-1}$, 87-m wind directions between 235° and 315°, and for normal turbine operation. Average deviation in NTF-corrected nacelle-mounted anemometer wind speed from tower 80-m wind speed is shown during stable conditions (blue) and during unstable conditions (red) versus tower 80-m wind speed with stability regimes based on (d) $R_B$, (e) $L$, and (f) $\alpha$. Dashed line indicates a 0 m s$^{-1}$ change.



**Table 3.** Coefficients for fifth-order polynomial NTFs for stability metrics.

| Regime | $R_B$ | | | $L$ | | |
|---|---|---|---|---|---|---|
| | Convective | Neutral | Stable | Convective | Neutral | Stable |
| $a_1$ | $-6.4141 \times 10^{-5}$ | $3.7809 \times 10^{-5}$ | $-3.0593 \times 10^{-4}$ | $-6.7085 \times 10^{-4}$ | $2.9071 \times 10^{-6}$ | $-3.8242 \times 10^{-5}$ |
| $a_2$ | 0.0030 | -0.0025 | 0.0142 | 0.0287 | $-4.4810 \times 10^{-4}$ | 0.0016 |
| $a_3$ | -0.0539 | 0.0621 | -0.2473 | -0.4721 | 0.0153 | -0.0245 |
| $a_4$ | 0.4628 | -0.6843 | 2.0334 | 3.7194 | -0.1881 | 0.1789 |
| $a_5$ | -0.8555 | 4.4391 | -6.8356 | -12.9316 | 2.0185 | 0.4534 |
| $a_6$ | 2.9265 | -5.9942 | 11.3853 | 19.7947 | -1.9273 | 0.5361 |

**Table 4.** Coefficients for fifth-order polynomial NTFs for the shear exponent.

| Regime | $\alpha$ | | |
|---|---|---|---|
| | Convective | Neutral | Stable |
| $a_1$ | $-2.3643 \times 10^{-5}$ | $1.4220 \times 10^{-5}$ | $-5.9409 \times 10^{-6}$ |
| $a_2$ | 0.0011 | -0.0011 | $7.3499 \times 10^{-5}$ |
| $a_3$ | -0.0202 | 0.0301 | 0.0038 |
| $a_4$ | 0.1781 | -0.3451 | -0.0689 |
| $a_5$ | 0.2889 | 2.7912 | 1.4373 |
| $a_6$ | 1.0387 | -3.2175 | -0.7869 |

This behavior shown by NTFs segregated by $R_B$ suggests that below rated speed in convective conditions, the nacelle anemometer underestimates the ambient wind speed more than in stable conditions. We speculate that at wind speeds below rated, mixing in the atmosphere during more convective conditions, as well as the turbine interaction with these turbulent eddies, may result in additional motion that exaggerates blockage effects by the rotor and nacelle and causes underestimation by the nacelle-mounted anemometer.




We apply the NTFs to the nacelle anemometer measurements to evaluate the deviations from the upwind
met tower data (Fig. 6d-f); however, the results show no consistency or systematic distinctions between stability
metrics, stability classes, or wind speed.

### 3.4 Turbulence effects on NTFs

The hypothesis that convectively-driven mixing and turbulence causes underestimation by the nacelle-mounted
anemometer is further supported in the NTFs segregated by TI (Fig. 7a, Table 5) and TKE classes (Fig. 7b, Table 5).
Distinctions between unstable and stable cases in the transfer functions for wind speeds between 5.5 and 12 m s$^{-1}$
are also apparent when the transfer functions are segregated by TI class (Fig. 7a) and for wind speeds around 12 m
s$^{-1}$ when the transfer functions are segregated by TKE class (Fig. 7b). Periods of relatively high TI and TKE result
in greater underestimations of the wind speed by the nacelle anemometer from just above cut-in wind speed to about
12 m s$^{-1}$.

Corrections to the nacelle wind speeds using NTFs based on atmospheric turbulence show lower deviations
from the ambient wind speed below rated speed and larger deviations from the ambient wind speed after rated speed
for high TI cases. However, similar to the results in Fig. 6d–f, Fig. 7c–d also show inconsistencies between
deviations from the upwind speed for the different turbulence metrics and regimes.





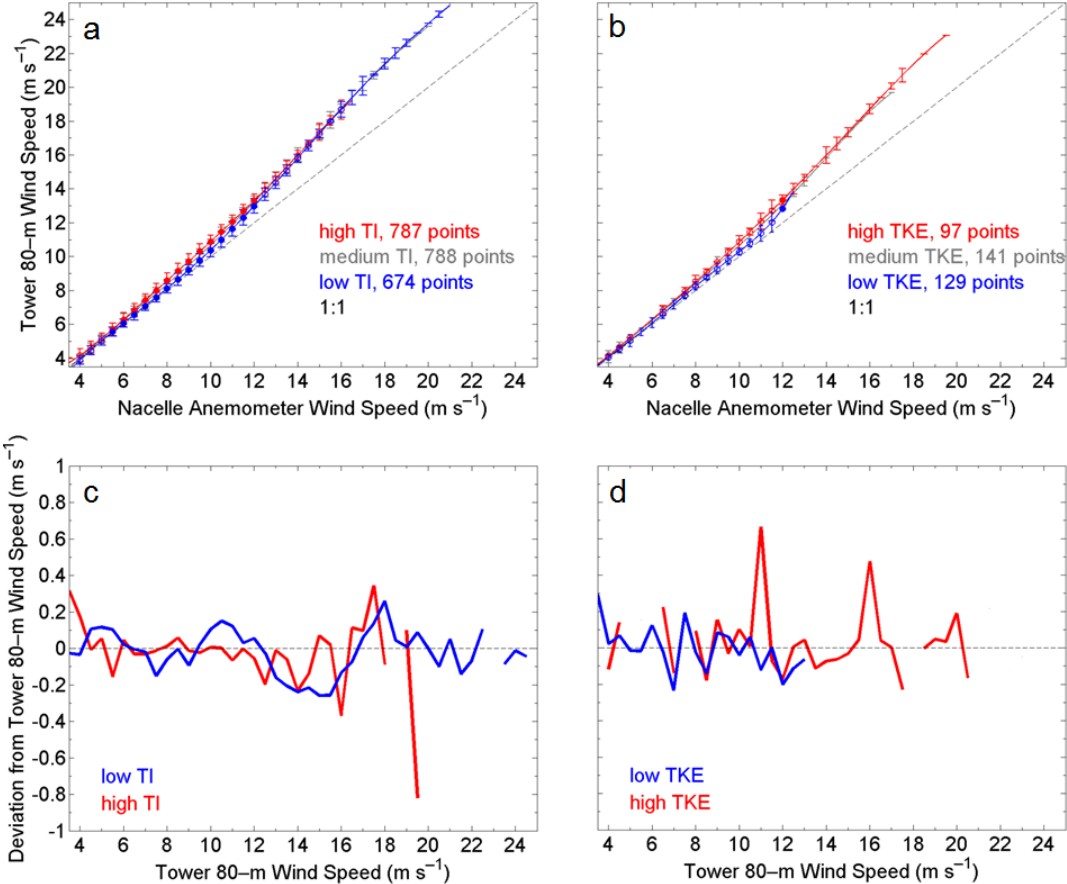

**Figure 7**. Tower 80-m NTFs calculated using fifth-order polynomial fits with turbulence regimes based on (a) turbulence intensity (TI) and (b) turbulence kinetic energy (TKE). Error bars represent $\pm \sigma$ of the data within the same bins as the bins with which the NTFs are calculated. Statistically distinct differences within each wind speed bin between the stability classes are denoted by closed circles. Figures include data filtered for tower 80-m wind speeds between 3.5 and 25.0 m s$^{-1}$, 87-m wind directions between 235° and 315°, and for normal turbine operation. Average deviation in NTF-corrected nacelle-mounted anemometer wind speed from tower 80-m wind speed is shown during stable conditions (blue) and during unstable conditions (red) versus tower 80-m wind speed with turbulence regimes based on (c) TI and (d) TKE. Dashed line indicates a 0 m s$^{-1}$ change.




**Table 5**. Coefficients for fifth-order polynomial NTFs for turbulence metrics.

| Regime | TI High | Med | Low | TKE High | Med | Low |
|---|---|---|---|---|---|---|
| $a_1$ | $-9.0463 \times 10^{-5}$ | $1.5534 \times 10^{-5}$ | $2.3161 \times 10^{-5}$ | $-1.3295 \times 10^{-5}$ | $-6.7464 \times 10^{-5}$ | $4.5266 \times 10^{-4}$ |
| $a_2$ | 0.0045 | -0.0012 | -0.0017 | $6.0813 \times 10^{-4}$ | 0.0031 | -0.0156 |
| $a_3$ | -0.0842 | 0.0305 | 0.0428 | -0.0103 | -0.0539 | 0.2048 |
| $a_4$ | 0.7602 | -0.3409 | -0.4763 | 0.0949 | 0.4418 | -1.2609 |
| $a_5$ | -2.1704 | 2.7552 | 3.3911 | 0.6268 | -0.6458 | 4.6488 |
| $a_6$ | 5.0077 | -3.2307 | -4.4045 | 0.6460 | 2.3183 | -3.9372 |

## 4 Conclusions

Over two months of data from both upwind instruments and nacelle-based instruments are used to quantify general

nacelle transfer functions (NTFs) as well as NTFs that vary with atmospheric stability and turbulence parameters.

We show that correcting nacelle winds using these NTFs results in more accurate annual energy production (AEP)

estimates that are similar to estimates obtained using upwind meteorological (met) tower-based wind speeds.

Further, multiple factors have been investigated for their influence on NTFs, including both parameters known to

influence wind power production and parameters never before investigated in the context of transfer functions.

We find that fitting the data to a fifth-order polynomial to estimate the NTF results in a slightly higher r-

squared value and smaller root-mean-square error (RMSE) than fitting to a second-order polynomial. The small

differences in the uncertainties between the two methods seem insignificant, as the r-squared value of 0.9909 using

the second-order polynomial is comparable to the 0.9912 value using the fifth-order fit. However, though the r-

squared value of the second-order fit is high, after correcting the nacelle winds with the second-order NTF, larger

deviations from the upwind tower winds occur than if a fifth-order NTF is used, especially at higher wind speeds.

At wind speeds below 9 m s$^{-1}$, the nacelle anemometer measurement closely corresponds to the upwind

wind speed measurement. Above this wind speed threshold, however, the nacelle anemometer underestimates the

upwind wind speed, which could result in a significant underestimation of power production and could be perceived

as turbine over-performance (or mask turbine under-performance) if not corrected for by a NTF. Additionally, the



non-linear nature of the transfer functions above 9 m s$^{-1}$ or so suggests that the transfer function may be impacted by

turbine operations near rated speed and how they affect the flow behind the rotor disk and along the nacelle.

The use of NTFs in AEP calculations results in a less than 1 % difference from the AEP calculated with the upwind met tower wind speed. AEP calculations reveal that an AEP calculated using a fifth-order polynomial correction to the nacelle winds results in a 0.003 % underestimation of the AEP calculated with the upwind wind speed, whereas an AEP calculated using a second-order polynomial correction results in a 0.18 % underestimation

of the AEP calculated with the upwind wind speed. Both are sizeable improvements over using the uncorrected nacelle wind speed, which leads to a 5.96 % overestimation when compared to the AEP calculated with the upwind wind speed.

Statistically significant distinctions emerge in the transfer functions for unstable and stable cases as defined by the Bulk Richardson number ($R_B$), particularly for wind speeds between 9 and 11 m s$^{-1}$. At these wind speeds

before rated, in unstable conditions, the nacelle anemometer underestimates the ambient wind speed more often than in stable conditions. Similar but more prominent behavior is found in transfer functions separated by turbulence intensity (TI) and turbulence kinetic energy (TKE) classifications: during periods with relatively high TI and TKE, the nacelle anemometer underestimates the ambient wind speed more than during periods of relatively low TI and TKE, between about 6 and 12 m s$^{-1}$. We speculate that turbine interaction with the mixing in the atmosphere during

more convective and turbulent conditions may result in additional motion, thereby exaggerating the blockage by the nacelle and thus underestimation by the nacelle-mounted anemometer.

Distinctions in power curves (Sumner and Masson, 2006; Antoniou et al., 2009; Vanderwende and Lundquist, 2012; Dörenkämper  et al., 2014; St. Martin et al., 2016) can lead to a correlation between these and distinctions in NTFs as well as the idea of validating power performance data with similar atmospheric and

operational characteristics with their corresponding power curve in an effort to decrease the amount of uncertainty in power performance testing.

NTFs have recently been accepted for power curve validation under certain circumstances (IEC 61400-12-2, 2013). They can also enable the use of nacelle-mounted anemometers for AEP estimates, turbine performance analysis, and data assimilation for improved forecasting (Draxl, 2012; Delle Monache et al., 2013).



Further work could explore how turbine controls and characteristics such as thrust affect the transfer functions. Simulations of flow around the nacelle such as those of Keck (2012) could be expanded to account for variations in atmospheric stability and could be coupled with control software simulators. As Bibor and Masson (2007) suggest, a single transfer function should not be used for every wind plant site and for every atmospheric and operating condition. Several atmospheric and operational conditions and how they affect the transfer functions

should be investigated and perhaps combined to provide an algorithm for manufacturers and wind plant operators to use in power performance validation.

**Acknowledgements**

The authors express appreciation to the Center for Research and Education in Wind for supporting this work, to Thomas Fischetti and Peter Gregg at GE Renewable Energy for their assistance in turbine data collection and

interpretation, and to the reviewers of a previous version of this work. This work was supported by the U.S. Department of Energy under Contract No. DE-AC36-08GO28308 with the National Renewable Energy Laboratory. Funding for the work was provided by the DOE Office of Energy Efficiency and Renewable Energy, Wind and Water Power Technologies Office.

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
