# Peer review of "Atmospheric turbulence affects wind turbine nacelle transfer functions"

_Wind Energy Science, 2016_

## Referee Comment (RC1) · Anonymous Referee #1 · 11 Jan 2017

General comments:

This manuscript quantitatively (statistically) analyzes the influence of different stability classes and turbulence regimes (obtained through the bulk Richardson number, turbulence intensity, etc.) on the wind turbine nacelle transfer functions. The authors analyzed data from one wind turbine and two sets of data from an upwind position from the wind turbine (mast and wind scanner data). The paper is well written and within the scope of Wind Energy Science.

The manuscript addresses an interesting subject that might have both practical and scientific applications in wind energy sector. However, the manuscript requires a number of clarifications throughout the text. Most of my questions are regarding the methodology and data, but I don't ask additional analyses to be conducted at this point. Namely,

it is not clear how the authors calculated some of the atmospheric quantities (e.g. bulk Richardson number). Interpretation of the results could also be better. Please see my specific comments below.

I recommend minor to moderate revisions for this manuscript before it can meet the publishing standards of Wind Energy Science.

Specific comments:

1. Anemometer and wind vane are not visible in Figure 1. The purpose of this figure (according to the text) is to show these instruments, but they are not visible. I advise the authors to add Figure 1b in which the anemometer and wind vane will be zoomed in (i.e. visible). The current Figure 1 can be Figure 1a.

2. The last paragraph in Introduction contains too many "as well as" phrases. Please reformulate these sentences in order to increase the readability of the text.

3. The last paragraph in Sect. 2.1 starts with "Further". I would suggest starting it with "Lastly."

4. Line 112. What do you mean by "simple, built-in transfer function" and how would this function modify the measured data? Please clarify as this might have importance for your results.

5. Lines 120-125. You estimated Weibull distribution parameters from the 2.5 months of data and then assumed that these parameters are representative for the whole year; am I right? Assuming that, you calculated the annual energy production. Can you please compare these calculated parameters against the parameters obtained from the data that actually cover one full year at that site, so we can see the uncertainty of your assumption and analysis?

6. Line 130. You are talking about near-surface flux measurements at 15 m and humidity measurements interpolated to 15 m, but in Sect. 2.1 (Meteorological and turbine data) you didn't mention any flux and/or relative humidity measurements. How/from

where did you obtain this data? Also, what kind of interpolation did you apply to get relative humidity at 15 m?

7. Similar to the previous comment, how did you calculate the virtual temperatures (absolute and potential) in order to obtain the bulk Richardson number values? That is, did you measure/calculate specific humidity or the mixing ration or the wet-bulb temperature? Please clarify.

8. I suggest you merge the last paragraph in Sect. 2.3 (Line 149) with the previous paragraph.

9. The caption for Fig. 3 can be simplified. You can say it's the same as Fig.2, but using second-order polynomial fit.

10. Line 220. If the nacelle anemometer underestimates the upwind winds, how is it possible that AEP based on the data from this anemometer is higher than using the upwind data? You provided an explanation, but I do not understand it. Please clarify.

11. The bottom row in Table 2 says "% difference from tower winds." If that's the name you choose, then the values are not accurately corresponding to that name. It indicates that AEP_upwind is 100% different from itself. Please simplify/rename and clarify.

12. The size of error bars and circles in Fig. 6 are not (very well) visible at 100% zoom. Please try to make these figures bigger as the interested reader is not able to actually estimate the errors from this graph.

13. The size and scaling of Fig 6. (bars, lines, points, etc.) are inadequate to develop the discussion that starts at Line 235 and ends at Line 247. Looking at Fig. 6a, I am not able to see any difference between the stable and unstable conditions and the arrows don't help much. Some discrepancies between the lines are visible at around 400% zoom.

14. Line 244. You believe that unstable conditions amplify the blockage effect and you carefully used the words "we speculate", "might be", "could be", etc., which I like.

[Figure]

However, can you provide some physical reasoning behind this speculation? Namely, why would the interaction between turbulent eddies and turbine augment the blockage effect and not diminish it? Your results show an augmentation (not very visible in Fig. 6 as it is now, but nevertheless show it), but what is the physics behind it?

15. References. Sometimes you used abbreviations for journal names and sometimes full names. Please be consistent.

---

## Referee Comment (RC2) · Anonymous Referee #2 · 12 Jan 2017

General comments:

This paper demonstrates a nacelle transfer function for "decontaminating" wind measurements mounted on the nacelle of an operating wind turbine. They also explore the impacts of thermal stability and turbulence regimes. The paper is fairly well written, but the Introduction and Data and Methods sections require some clarification, and would benefit from concision.

I am not entirely convinced of the practical application of this technique. Your technique requires contemporaneous measurements from an "upwind" tower, but in practice such measurements often are not available. You even acknowledge this in the introduction:

"However, it is not feasible to erect "site calibration" met towers after the turbine has been erected. And, even if "site calibration" is not required because a site is in simple

terrain, tower erection is time consuming and unrealistic to complete for every turbine at a given park."

Perhaps I am missing important details, but I do not understand how this technique could be applied in the absence of an upwind measurement(s). And those measurements need to be representative of the site. In regions such as Europe, these kind measurements are exceeding rare at operating projects, and it is not clear how applicable this approach is in practice.

Specific comments:

(1) There is insufficient information about the methods and rationale. The reader is frequently referred other papers for these important details. For example, lines 149-151 of the paper state that: "Regimes or classifications for these stability and turbulence parameters are defined in Table 1 and described in detail in St. Martin et al. (2016), along with more detailed descriptions of the data from the lidar, tower and turbine, as well as filtering methods."

A scientific paper should be entirely self-contained, and provide enough information for the reader to readily understand what you have done and how you have done it. We should not be forced to locate and dig through other papers for the details of your methods.

(2) The classifications in Table 1 seem arbitrary, particularly for the TI and TKE "high", "medium", and "low". Without context and and understanding of how you arrived at these classifications, they seem very subjective.

(3) There are a number of confusion passages in the Introduction and Data and Methods sections. For example, the paragraph starting on line 58 is very hard to follow, and could be greatly shortened without losing the salient information. Here is my humble attempt, which combines the two paragraphs spanning lines 57-77):

"The relationship between UHWS measurements and NAWS measurements used for

generating NTFs has been found to depend on a number of factors, including: nacelle height, wind inflow angle, blade pitch angle, yaw misalignment, the position of the anemometer on the nacelle, the anemometer calibration, and the characteristics of the surrounding terrain (References .... ). However, the impacts of inflow turbulence and atmospheric stability on NTFs have not yet been explored, even though it has been recognized that they may play an important role (References ....)."

(4) Lines 95 and 96: Change "(2.7 D upwind)" and "(2.0 D upwind)" to "(2.7 rotor diameters upwind; AND STATE THE PHYSICAL DISTANCE!)" and "(2.0 rotor diameters upwind)".

(5) Lines 100-104: This is really hard to follow, and keep the figures straight. I strongly suggest that you put this into a Table, which will be much easier to digest. This is also one of many places you refer the reader to some other paper for more details–in this case the configuration of met tower. Very frustrating!

---

## Author Response (AR1)

**Response to Reviewer 1 comments:**

*General comments: This manuscript quantitatively (statistically) analyzes the influence of different stability classes and turbulence regimes (obtained through the bulk Richardson number, turbulence intensity, etc.) on the wind turbine nacelle transfer functions. The authors analyzed data from one wind turbine and two sets of data from an upwind position from the wind turbine (mast and wind scanner data). The paper is well written and within the scope of Wind Energy Science.*

The authors thank the reviewer for their kind comments.

*The manuscript addresses an interesting subject that might have both practical and scientific applications in wind energy sector. However, the manuscript requires a number of clarifications throughout the text. Most of my questions are regarding the methodology and data, but I don't ask additional analyses to be conducted at this point. Namely, it is not clear how the authors calculated some of the atmospheric quantities (e.g. bulk Richardson number). Interpretation of the results could also be better. Please see my specific comments below.*

*I recommend minor to moderate revisions for this manuscript before it can meet the publishing standards of Wind Energy Science.*

*Specific comments:*

*1. Anemometer and wind vane are not visible in Figure 1. The purpose of this figure (according to the text) is to show these instruments, but they are not visible. I advise the authors to add Figure 1b in which the anemometer and wind vane will be zoomed in (i.e. visible). The current Figure 1 can be Figure 1a.*

We thank the reviewer for pointing this out, and we have found another picture in which the nacelle anemometer is more visible. We have replaced the current Figure 1 with the following:

[Figure]

**Figure 1**. GE-1.5/77 sle turbine at the National Wind Technology Center. Photo credit: Dennis Schroeder/NREL (image gallery number 29611).

*2. The last paragraph in Introduction contains too many "as well as" phrases. Please reformulate these sentences in order to increase the readability of the text.*

We have changed to the following (changes are represented by the bold text):

"In this study, we quantify the effect of NTF-corrected nacelle anemometer measurements on the AEP and investigate the influence of different atmospheric stability and turbulence regimes on these NTFs. In Sect. 2, we briefly summarize our data set, which includes upwind **and** nacelle-based measurements, **as well as** our data analysis methods which include filtering based on turbine operation, and definitions of the stability and turbulence regimes. We present results of AEP calculations **together with results of** separate NTFs for different stability and turbulence regimes in Sect. 3**. In** Sect. 4 we summarize conclusions about the effect of the NTF on the AEP **in addition to** the effects of atmospheric stability and inflow turbulence on the NTFs."

*3. The last paragraph in Sect. 2.1 starts with "Further". I would suggest starting it with "Lastly."*

We thank the reviewer for the suggestion and we have changed to "Lastly."

*4. Line 112. What do you mean by "simple, built-in transfer function" and how would this function modify the measured data? Please clarify as this might have importance for your results.*

We agree that this statement is not clear, and have revised the paragraph as such:

"Lastly, the nacelle-reported wind speeds used in this analysis have been subjected to a simple, linear regression transfer function before the retrieval from the SCADA system of the DOE GE 1.5 sle turbine. This linear regression function, built into the SCADA system by the turbine manufacturer, effectively translates the raw signal from the cup anemometer to wind speed and is not unlike a transfer function provided by an anemometer manufacturer. We see the uncertainty of this built-in transfer function as an advantage to our analysis as a typical wind plant operator would only have access to similar data."

*5. Lines 120-125. You estimated Weibull distribution parameters from the 2.5 months of data and then assumed that these parameters are representative for the whole year; am I right? Assuming that, you calculated the annual energy production. Can you please compare these calculated parameters against the parameters obtained from the data that actually cover one full year at that site, so we can see the uncertainty of your assumption and analysis?*

Yes, we calculated Weibull parameters based on the 2.5 months of hub-height wind speed data, filtered by wind direction sector, wind speed, and curtailment. These Weibull parameters were then used to calculate sample wind distributions for the AEP estimates. We then extrapolate the AEP estimates to one year so the AEP values are more characteristic of typical AEP numbers at sites suitable for wind development. We have clarified this reasoning and have added the following in bold to the end of Sect. 2.2:

"A sample wind distribution using Weibull distribution parameters representative of the data set (scale parameter: $\lambda = 10.04$ m s$^{-1}$, shape parameter: $k = 2.63$, figure not shown) is used in these calculations as suggested by IEC 61400 12-1 (2015) for a site-specific AEP. **We note that these parameters, based on 2.5 months in the "high" wind season at this site, are not actually representative of the entire year. However, as noted in other analyses of this test site (Clifton and Lundquist, 2012; Clifton et al., 2013), this site would not be chosen for wind development given the long summer season with little or no wind. We emphasize that this approach is not meant to suggest actual AEPs for this site, but to explore the sensitivity of AEP calculations at sites reasonable for wind development.** "

Because of the additional text, the following references have been added:

Clifton,A.and Lundquist,J.K.:Data clustering reveals climate impacts on local phenomena, J. Appl. Meteorol. Clim., 51, 1547– 1557, doi:10.1175/JAMC-D-11-0227.1, 2012.

Clifton, A., Schreck, S., Scott, G., and Lundquist, J. K.: Turbine inflow characterization at the National Wind Technology Center, J. Sol. Energ.-T. ASME, 135, 031017, doi:10.1115/1.4024068, 2013.

*6. Line 130. You are talking about near-surface flux measurements at 15 m and humidity measurements interpolated to 15m, but in Sect. 2.1 (Meteorological and turbine data) you didn't mention any flux and/or relative humidity measurements. How/from where did you obtain this data? Also, what kind of interpolation did you apply to get relative humidity at 15 m?*

A 3-D sonic anemometer mounted at 15 m on the tower as described in Sect. 2.1 provides measurements of the vertical component of the wind as well as sonic temperature. To make this clear, we have added the following in bold:

"On the met tower, cup anemometers placed at 3, 10, 30, 38, 55, 80, 87, 105, 122, and 130 m measure wind speed and vanes placed at 3, 10, 38, 87, and 122 m measure wind direction. Three-dimensional (3-D) sonic anemometers placed at 15, 41, 61, 74, 100, and 119 m **measure all three components of the wind as well as sonic temperature which are used to calculate momentum and heat fluxes**."

As for humidity measurements, we have added the following in bold:

"Barometric pressure and precipitation amounts are measured at 3 m, temperature is measured at 3, 38, and 87 m **and dew point temperature is measured at 3, 38, 87, and 122 m**."

As well as:

Using near-surface flux measurements at 15 m (within the surface layer) as well as surface temperature and humidity measurements **linearly** interpolated to 15 m, we calculate 30-min values of L to estimate the height at which the buoyant production of turbulence dominates the mechanical production of turbulence."

We have also added the following table to make it easier to visualize the tower configuration:

| Instrument | Mounting heights (m) |
|---|---|
| Cup anemometer | 3, 10, 30, 38, 55, 80, 87, 105, 122, 130 |
| Wind vane | 3, 10, 38, 87, 122 |
| 3-D sonic anemometer | 15, 41, 61, 74, 100, 119 |
| Barometric pressure sensor | 3 |
| Precipitation sensor | 3 |
| Temperature sensor | 3, 38, 87 |
| Dew point temperature sensor | 3, 38, 87, 122 |

*7. Similar to the previous comment, how did you calculate the virtual temperatures (absolute and potential) in order to obtain the bulk Richardson number values? That is, did you*

*measure/calculate specific humidity or the mixing ration or the wet-bulb temperature? Please clarify.*

We thank the reviewer for pointing out that we did not explain in full our Bulk Richardson number calculations. We have added the following equation and reference to Stull (1988):

$$R_B = \frac{g \Delta T \Delta z}{\overline{T} \Delta U^2}$$

where g is the gravitational constant, $\Delta T$ is the change is temperature across $\Delta z$, $\overline{T}$ is the mean temperature across $\Delta z$, and $\Delta U$ is the change in horizontal wind speed across $\Delta z$. Humidity is not considered in this formulation of the bulk Richardson number.

*8. I suggest you merge the last paragraph in Sect. 2.3 (Line 149) with the previous paragraph.*

We thank the reviewer for this suggestion and have merged the last paragraph in Sect. 2.3 with the previous paragraph.

*9. The caption for Fig. 3 can be simplified. You can say it's the same as Fig.2, but using second-order polynomial fit.*

We have simplified the caption for Fig. 3 from:

"Comparison of upwind wind speeds with nacelle anemometer wind speeds. (a) Scatter is the upwind tower 80-m wind speed versus nacelle wind speed. Red line is the second-order polynomial fit and empirical transfer function between the tower 80-m observations and the nacelle-mounted anemometer observations; gray line is the fifth-order polynomial fit. Dashed line is 1:1. (b) Average deviation in the second-order polynomial NTF-corrected nacelle-mounted anemometer wind speed from tower 80-m wind speed versus tower 80-m wind speed is shown. Dashed line indicates a 0 m s$^{-1}$ change. Includes data filtered for tower 80-m wind speeds between 3.5 and 25.0 m s$^{-1}$, 87-m wind directions between 235° and 315°, and for normal turbine operation."

to:

"Comparison of upwind wind speeds with nacelle anemometer wind speeds. The red line in (a) is the second-order polynomial fit and empirical transfer function between the tower 80-m observations and the nacelle-mounted anemometer observations and the gray line in (a) is the fifth-order polynomial fit. (b) Average deviation in the second-order polynomial NTF-corrected nacelle-mounted anemometer wind speed from tower 80-m wind speed versus tower 80-m wind speed is shown."

*10. Line 220. If the nacelle anemometer underestimates the upwind winds, how is it possible that AEP based on the data from this anemometer is higher than using the upwind data? You provided an explanation, but I do not understand it. Please clarify.*

If the nacelle anemometer underestimates the wind speed, the power curve is essentially shifted to the left. This is because the turbine appears to produce more power at lower wind speeds. For example, if the turbine is producing 400 kW, the corresponding wind speed according to the manufacturer power curve is 7 m s$^{-1}$, however, the anemometer may be reading 6 m s$^{-1}$, so it seems like the turbine is producing more power at lower wind speeds, moving the power curve over and increasing the AEP. Additional text indicated by the bold font has been added to the following to make this more clear:

"This overestimation of AEP is expected as the nacelle anemometer consistently underestimates the upwind wind speed, which leads to the misrepresentation of higher power output at lower wind speeds**, effectively shifting the entire power curve to the left**, and therefore **leading to** a higher AEP."

*11. The bottom row in Table 2 says "% difference from tower winds." If that's the name you choose, then the values are not accurately corresponding to that name. It indicates that AEP_upwind is 100% different from itself. Please simplify/rename and clarify.*

Thank you for pointing this out, and we have corrected the label to "% of tower winds".

*12. The size of error bars and circles in Fig. 6 are not (very well) visible at 100% zoom. Please try to make these figures bigger as the interested reader is not able to actually estimate the errors from this graph.*

Based on your comment, we have decided to split this figure up to enlarge the individual panels. So we have taken Fig. 6 and shown (a) and (d) as one figure, (b) and (e) as another, and (c) and (f) as a third figure. This way the reader will be able to more clearly see bins where regimes are statistically distinct from one another. Example:

[Figure]

*13. The size and scaling of Fig 6. (bars, lines, points, etc.) are inadequate to develop the discussion that starts at Line 235 and ends at Line 247. Looking at Fig. 6a, I am not able to see*

*any difference between the stable and unstable conditions and the arrows don't help much. Some discrepancies between the lines are visible at around 400% zoom.*

Our answer to the above comment will make it easier for the reader to see these statistical differences. In addition, we have moved the discussion starting at line 243 to a new section after showing turbulence results in an additional section 3.5 titled "Discussion", in which we have expanded to include more explanation on a physical reasoning behind these results.

*14. Line 244. You believe that unstable conditions amplify the blockage effect and you carefully used the words "we speculate", "might be", "could be", etc., which I like. However, can you provide some physical reasoning behind this speculation? Namely, why would the interaction between turbulent eddies and turbine augment the blockage effect and not diminish it? Your results show an augmentation (not very visible in Fig. 6 as it is now, but nevertheless show it), but what is the physics behind it?*

We have added the following in bold and moved to a new discussion section 3.5:

"We speculate that at wind speeds below rated, mixing in the atmosphere during more convective conditions, as well as the turbine interaction with these turbulent eddies, may result in additional motion that exaggerates blockage effects by the rotor and nacelle and causes underestimation by the nacelle-mounted anemometer. We suspect that rotor response is lagging in more convective and turbulent conditions as the turbine responds more quickly to drops in wind speed. Therefore, during more turbulent conditions, it is possible that lower rotor efficiency influences flow induction and thus the wind speeds measured on the back of the nacelle. If turbine and rotor efficiencies are lower during periods with convective and more turbulent conditions, it may be surmised then, that less momentum passes through the rotor and along the nacelle. In addition, power curve results from the same dataset discussed here (St. Martin et al., 2016) show that during less stable and more turbulent conditions at wind speeds within the ramp-up region of the power curve, more power is produced than during periods of more stable and less turbulent conditions. Power production will also affect the flow induction (Frandsen et al., 2009) and thus the wind speed directly behind the rotor disk: if more energy is extracted by the rotor, the nacelle-mounted anemometer will likely measure lower winds."

*15. References. Sometimes you used abbreviations for journal names and sometimes full names. Please be consistent.*

We thank the reviewer for pointing this out and have made sure we use journal abbreviations consistently throughout our reference list.

**Response to Reviewer 2 comments:**

*General comments: This paper demonstrates a nacelle transfer function for "decontaminating" wind measurements mounted on the nacelle of an operating wind turbine. They also explore the impacts of thermal stability and turbulence regimes. The paper is fairly well written, but the Introduction and Data and Methods sections require some clarification, and would benefit from concision.*

*I am not entirely convinced of the practical application of this technique. Your technique requires contemporaneous measurements from an "upwind" tower, but in practice such measurements often are not available. You even acknowledge this in the introduction: "However, it is not feasible to erect "site calibration" met towers after the turbine has been erected. And, even if "site calibration" is not required because a site is in simple terrain, tower erection is time consuming and unrealistic to complete for every turbine at a given park." Perhaps I am missing important details, but I do not understand how this technique could be applied in the absence of an upwind measurement(s). And those measurements need to be representative of the site. In regions such as Europe, these kind measurements are exceeding rare at operating projects, and it is not clear how applicable this approach is in practice.*

As stated in Sect. 1:

"Nacelle measurements could also be used to help improve turbine or park efficiency. For example, power performance verifications for individual turbines can now be based on the nacelle anemometer with suitable nacelle transfer functions (NTFs) (International Electrotechnical Commission [IEC] 61400-12-2 2013). Nacelle measurements can also provide critical input for wind farm production optimization (Fleming et al., 2016). With sufficiently accurate NTFs, these data can provide a valuable, extensive, and continuous source of turbine-specific performance information."

The analysis presented in this work is motivated by the application of power performance testing as stated in the above excerpt. Typically, power performance testing is performed using measurements from some upwind tower or remote sensing instrument, however, IEC standards released in 2013 (IEC 61400-12-2) titled "Power performance of electricity-producing wind turbines based on nacelle anemometry" describe how nacelle anemometer measurements can be used for this application if based on transfer functions. Quantifying these transfer functions require upwind measurements be available at some point post-construction. However, once transfer functions are calculated for a site, the tower can be taken down and the transfer functions used to correct the nacelle measurements for future performance testing. The IEC standards (IEC 61400-12-2, 2013) even allow the use of these transfer functions at other, similar sites as stated within the standard:

"The procedure can be used for power performance evaluation of specific turbines at specific locations, but equally the methodology can be used to make generic comparisons between different turbine models or different turbine settings."

…

"If during an NPC [nacelle power curve] measurement an NTF is used that has previously been measured in another park it may only be applied to turbines with a terrain classification that is the same as the terrain class during the NTF measurements; the terrain must also have the same sign of terrain slope in the measurement sector."

This work focuses on the effect of atmospheric conditions on these transfer functions, so that when operators perform these calculations for future power performance testing, they will be aware of some of the factors these transfer functions are sensitive to. The manuscript ends with the following:

"Several atmospheric and operational conditions and how they affect the transfer functions should be investigated and perhaps combined to provide an algorithm for manufacturers and wind plant operators to use in power performance validation."

To further clarify the practicality of the NTF, we have added the following to Sect. 1 when introducing NTFs:

"Quantifying these transfer functions require upwind measurements be available at some point post-construction. However, once transfer functions are calculated for a site, the tower can be taken down and the transfer functions used to correct the nacelle measurements for future performance testing."

*Specific comments:*

*(1) There is insufficient information about the methods and rationale. The reader is frequently referred other papers for these important details. For example, lines 149-151 of the paper state that: "Regimes or classifications for these stability and turbulence parameters are defined in Table 1 and described in detail in St. Martin et al. (2016), along with more detailed descriptions of the data from the lidar, tower and turbine, as well as filtering methods." A scientific paper should be entirely self-contained, and provide enough information for the reader to readily understand what you have done and how you have done it. We should not be forced to locate and dig through other papers for the details of your methods. (2) The classifications in Table 1 seem arbitrary, particularly for the TI and TKE "high", "medium", and "low". Without context and and understanding of how you arrived at these classifications, they seem very subjective.*

We understand, and have added equations for each stability metric with definitions of each parameter as well as the following:

"Regimes of TI, TKE, and α are determined by splitting the distributions of each parameter roughly into thirds. Regimes of $R_B$ are similarly determined, as in Aitken et al. (2014) and St. Martin et al. (2016), and uncertainty in the $R_B$ values calculated from propagation of instrument accuracy ensures the regimes are wide enough. Stability regimes based on $L$ are similar to those defined by Muñoz-Esparza (2012)."

Because of the additional text, the following reference has been added:

Muñoz-Esparza, D., Cañadillas, B., Neumann, T., and vanBeech, J.: Turbulent fluxes, stability and shear in the offshore environment: mesoscale modelling and field observations at FINO1, J. Renew. Sustain. Energy, 4, 1–16, doi:10.1063/1.4769201, 2012.

*(3) There are a number of confusion passages in the Introduction and Data and Methods sections. For example, the paragraph starting on line 58 is very hard to follow, and could be greatly shortened without losing the salient information. Here is my humble attempt, which combines the two paragraphs spanning lines 57-77):*

*"The relationship between UHWS measurements and NAWS measurements used for generating NTFs has been found to depend on a number of factors, including: nacelle height, wind inflow angle, blade pitch angle, yaw misalignment, the position of the anemometer on the nacelle, the anemometer calibration, and the characteristics of the surrounding terrain (References .... ). However, the impacts of inflow turbulence and atmospheric stability on NTFs have not yet been explored, even though it has been recognized that they may play an important role (References ....)."*

Thank you for the suggestion. To be more concise, we have changed the text in the Introduction from:

"In previous work, the relationship between UHWS measurements and NAWS measurements has been found to depend on multiple factors. Antoniou and Pedersen (1997) found that relations between the UHWS and the NAWS were dependent on rotor settings such as blade pitch angle and the use of vortex generators, yaw error, anemometer position, and terrain. They concluded that if these factors were kept constant, the relation could be used for all wind turbines of the same make and type. Frandsen et al. (2009) found a dependence on flow induction caused by the rotor. Dahlberg et al. (1999) discovered that pitch angle affects the relation. Dahlberg et al. (1999), Smith et al. (2002), and Frandsen et al. (2009) also stressed the importance of the correct calibration of the nacelle anemometers and that this calibration has an effect on the error measured in the relation. Zahle and Sørensen (2011) found that the inflow angle to the rotor and yaw misalignment influences the relationship. Smith et al. (2002) concluded the relation may depend on turbine controls, topography, and nacelle height and position. Smaïli and Masson (2004) implemented a numerical model and concluded that a relation should account for rotor-nacelle interactions and hypothesized that wakes, topography, and nacelle misalignment would all have some effect on the relation. To summarize, the factors found to be relevant in NTFs are:

rotor settings, yaw error, anemometer position, terrain, flow induction (decrease in wind speed just in front of or just behind the rotor), nacelle anemometer calibration, and inflow angle."

To:

"In previous work, the relationship between UHWS measurements and NAWS measurements has been found to depend on multiple factors, including rotor and turbine control settings such as blade pitch angle and inflow angle, the use of vortex generators, yaw error, terrain, flow induction, calibration of the anemometer, and nacelle height and position (Antoniou and Pedersen, 1997; Dahlberg et al., 1999; Smith et al., 2002; Smaïli and Masson, 2004; Frandsen et al., 2009; Zahle and Sørensen, 2011)."

*(4) Lines 95 and 96: Change "(2.7 D upwind)" and "(2.0 D upwind)" to "(2.7 rotor diameters upwind; AND STATE THE PHYSICAL DISTANCE!)" and "(2.0 rotor diameters upwind)".*

We agree and have made the following changes in bold:

"Upwind data include 1-Hz measurements of wind speed and direction averaged to 10 min from a Renewable NRG Systems (NRG)/LEOSPHERE WINDCUBE v1 vertically profiling Doppler lidar (2.7 **rotor diameters (D) upwind; 208 m** ) and 10- and 30-min averages from a 135-m met tower (2.0 D upwind, **154 m**)."

*(5) Lines 100-104: This is really hard to follow, and keep the figures straight. I strongly suggest that you put this into a Table, which will be much easier to digest. This is also one of many places you refer the reader to some other paper for more details–in this case the configuration of met tower. Very frustrating!*

We have placed the following table to make it easier for the reader to absorb and retain the information:

| Instrument | Mounting heights (m) |
|---|---|
| Cup anemometer | 3, 10, 30, 38, 55, 80, 87, 105, 122, 130 |
| Wind vane | 3, 10, 38, 87, 122 |
| 3-D sonic anemometer | 15, 41, 61, 74, 100, 119 |
| Barometric pressure sensor | 3 |
| Precipitation sensor | 3 |
| Temperature sensor | 3, 38, 87 |
| Dew point temperature sensor | 3, 38, 87, 122 |

**Atmospheric turbulence affects wind turbine nacelle transfer functions**

Clara M. St. Martin,[1] Julie K. Lundquist,[1,2] Andrew Clifton,[2] Gregory S. Poulos,[3] and Scott J.

Schreck[2]

[1] Department of Atmospheric and Oceanic Sciences (ATOC), University of Colorado at Boulder, 311 UCB, Boulder, CO, 80309

[2] National Renewable Energy Laboratory, 15013 Denver West Parkway, Golden, CO, 80401

[3] V-Bar, LLC, 1301 Arapahoe Street, Suite 105, Golden, CO, 80401

Correspondence to: Clara M. St. Martin (clara.st.martin@colorado.edu)

**Abstract.** Despite their potential as a valuable source of individual turbine power performance and turbine array energy production optimization information, nacelle-mounted anemometers have often been neglected because complex flows around the blades and nacelle interfere with their measurements. This work quantitatively explores the accuracy of and potential corrections to nacelle anemometer measurements to determine the degree to which they may be useful when corrected for these complex flows, particularly for calculating annual energy production (AEP) in the absence of other meteorological data. Using upwind meteorological tower measurements along with nacelle-based measurements from a General Electric (GE) 1.5sle model, we calculate empirical nacelle transfer functions (NTFs) and explore how they are impacted by different atmospheric and turbulence parameters. This work provides guidelines for the use of NTFs for deriving useful wind measurements from nacelle-mounted anemometers. Corrections to the nacelle anemometer wind speed measurements can be made with NTFs and used to calculate an AEP that comes within 1 % of an AEP calculated with upwind measurements. We also calculate unique NTFs for different atmospheric conditions defined by temperature stratification as well as turbulence intensity, turbulence kinetic energy, and wind shear. During periods of low stability as defined by the Bulk Richardson number ($R_B$), the nacelle-mounted anemometer underestimates the upwind wind speed more than during periods of high stability at some wind speed bins below rated speed, leading to a more steep NTF during periods of low stability. Similarly, during periods of high turbulence, the nacelle-mounted anemometer underestimates the upwind wind speed more than during periods of low turbulence at most wind bins between cut-in and rated wind speed. Based on these results, we suggest different NTFs be calculated for different regimes of atmospheric stability and turbulence for power performance validation purposes.

**Keywords**

nacelle anemometry, nacelle transfer function, atmospheric stability, turbulence

**Copyright statement**

The U.S. Government retains and the publisher, by accepting the article for publication, acknowledges that the U.S. Government retains a nonexclusive, paid-up, irrevocable, worldwide license to publish or reproduce the published form of this work, or allow others to do so, for U.S. Government purposes.

[revised manuscript text omitted]